# The Role of Platelet-Derived Extracellular Vesicles in Immune-Mediated Thrombosis

**DOI:** 10.3390/ijms23147837

**Published:** 2022-07-16

**Authors:** Alicia S. Eustes, Sanjana Dayal

**Affiliations:** 1Department of Internal Medicine, Carver College of Medicine, University of Iowa, Iowa City, IA 52242, USA; alicia-eustes@uiowa.edu; 2Iowa City VA Healthcare System, Iowa City, IA 52246, USA

**Keywords:** platelet-derived extracellular vesicles, thrombosis, immune mediated

## Abstract

Platelet-derived extracellular vesicles (PEVs) play important roles in hemostasis and thrombosis. There are three major types of PEVs described based on their size and characteristics, but newer types may continue to emerge owing to the ongoing improvement in the methodologies and terms used to define various types of EVs. As the literature on EVs is growing, there are continuing attempts to standardize protocols for EV isolation and reach consensus in the field. This review provides information on mechanisms of PEV production, characteristics, cellular interaction, and their pathological role, especially in autoimmune and infectious diseases. We also highlight the mechanisms through which PEVs can activate parent cells in a feedback loop.

## 1. Introduction

In 1967, extracellular vesicles (EVs) were first discovered [1]. Initially, these vesicles were considered “platelet dust” and were found to influence thrombosis and hemostasis but have since been found to play roles in many different conditions, including in inflammation and immunologic responses. The majority of these EVs are derived from platelets [2,3,4,5,6,7,8,9,10]. The exact percentage that platelet-derived EVs account for is controversial and varies from 30–85%, with most studies reporting higher percentages [2,3,4,5,6,7,8,9,10].

An ever-increasing number of reports are being published on the function and potential therapeutic roles of EVs. However, there have been inconsistencies in the definitions of EVs and their isolation methods. To address this issue, the International Society for Extracellular Vesicles (ISEV) enacted the Minimal Information for Studies of Extracellular Vesicles (MISEV) guidelines in 2014, which were updated in 2018 [11]. These guidelines are intended to be used for mouse and human EVs. In these guidelines, EV is classified as the generic term used for vesicles that are released from a cell, have a lipid bilayer, and cannot replicate. There are multiple populations of EVs that can be characterized further by their size, contents, and formulation. However, with technological advances, new populations of EVs can and are being discovered. Some studies have found EVs the size of exosomes, or smaller, that are made through different mechanisms than those mentioned below [12,13] and contain different contents [14] or are even non-membranous [15] (Figure 1). As technology continues to advance, we will undoubtedly see the emergence of different EV populations or subgroups. For now, the classic populations are exosomes, ectosomes (microparticles/microvesicles), and apoptotic bodies.

## 2. Type of EVs and Their Characterization

### 2.1. Exosomes

Exosomes are small EVs that are 30–150 nm in diameter [16,17,18] (Figure 1). Typically, exosomes are less heterogenous in size and content than other EVs because they are produced and secreted through a regulated process. In the first step of this process, there is an invagination of endosomes in a ceramide-regulated fashion [19], resulting in the formation of multivesicular bodies (MVBs). While the MVB is critical in exosome formation for most cells, platelet-derived exosomes can originate from MVBs or from alpha granules [20,21] (Figure 2). This process is dependent on the pH of the endosomes. Lower pH destines endosomes for the lysosomal degradation pathway, while a higher pH moves the endosomes towards the plasma membrane and results in an increased exosome yield [16,22] (Figure 2). The acidification of the endosome is reliant on vacuolar H^+^-ATPase activity. Cholesterol accumulation in the MVB, activating vacuolar H^+^-ATPase, results in endosomal acidification and fewer exosomes are produced. On the other hand, neutral sphingomyelinase 2, which inhibits vacuolar H^+^-ATPase activity, results in increased exosome production [22]. Once the MVB has formed, the MVB membrane will then fuse with the plasma membrane, releasing the exosomes. The endosomal sorting complex required for transport (ESCRT) proteins is heavily involved in the fusion of the MVB and plasma membrane, and as such essential for the secretion of exosomes [23]. Many other proteins associated with the cytoskeleton are involved with exosome secretion as well. One such family of proteins are the Rabs [16]. Signaling from outside of the cell that affects these processes can also affect exosome production. For example, syndecan proteoglycans interact with ALG-2 interacting protein X (ALIX) which then interacts with ESCRT proteins and affects exosome numbers [24]. As another example, tumor necrosis factor-alpha (TNFα) can activate neutral sphingomyelinase 2 which then results in increased exosome production [22].

Because the formation of exosomes is a regulated process, the packaging of RNAs, lipids, and proteins into the exosomes also occurs in a regulated manner. There are still many unknowns as to how this process is regulated, but it has been found that components of the secretory pathway are involved in this regulation [25]. Furthermore, post-translational modification of proteins, such as ubiquitination and glycosylation, can regulate their incorporation into exosomes [26,27,28,29,30,31,32]. The effect post-translational modifications have on cargo sorting depends on the type of modification. While specifics may vary depending on the protein or exact modification, in general, ubiquitination, glycosylation, and sumoylation increase target packaging into exosomes, while acetylation and ISGylation target the protein for degradation instead.

### 2.2. Microparticles/Microvesicles

Ectosomes (microparticles/microvesicles) are EVs of 150–1000 nm in diameter (Figure 1) and are formed through blebbing of the plasma membrane. The production of ectosomes is not a regulated process, and thus is not fully characterized. However, there is evidence that the actin cytoskeleton is important in this process. Benedikter et al. found that ectosome production is regulated by the generation of reactive-oxygen species (ROS) [33]. ROS act on actin to induce actin polymerization, and one mechanism is through ROS-induced S-nitrosylation of cytosolic actin that results in enhanced actin polymerization. This in turn leads to increased ectosome production [34]. Platelet-derived ectosome release requires intact actin dynamics while tubulin is dispensable [35]. The use of cytochalasin D, an actin polymerization inhibitor, inhibits platelet-derived ectosome formation [36]. Conversely, while megakaryocyte-derived ectosome production is also independent of tubulin, it is augmented by actin polymerization inhibition [37]. This opposing phenotype is potentially related to the critical requirement of actin in megakaryopoiesis. Interestingly, the addition of the phosphatase inhibitors, okadaic acid and calyculin A, both of which increase cytoskeletal rearrangement, results in doubled platelet-derived ectosome formation [36].

There are three possible steps in which the actin cytoskeleton regulates ectosome formation [38]. First, the force generated from the polymerization of the actin could lead to bud formation in the plasma membrane. Second, the myosin mechanoenzymes in the cell could transport the components to the site of ectosome production. Third, the actin-myosin network could act to close and separate the ectosome from the host cell. Likely all three are involved. Filamin A promotes the branching of actin and anchors transmembrane proteins and glycoproteins to the actin cytoskeleton. Collier et al. discovered that while a partial decrease in filamin A expression had no effect on ectosome size or number, a 90% decrease in expression lead to increased ectosome release from the cell [39]. In particular, Collier et al. found that filamin A was involved in the regulation of tissue factor (TF) incorporation into procoagulant ectosomes [39,40]. They hypothesized that this was due to increased instability of the cytoskeleton at the plasma membrane surface. Similarly, others have found that disruption of the actin cytoskeleton in platelets, through calpain or integrin activation, leads to increased ectosome formation [41,42,43]. Calpain activity, resulting in the cleavage of cytoskeletal proteins, correlated with platelet-derived ectosome formation, and the inhibition of calpain blocked ectosome formation [41,42,44]. Intriguingly, the role of calpain appeared to be dependent on the stimulant used. When platelets were stimulated with calcium ionophore A23187, calpain inhibition depleted ectosome formation [44]. On the other hand, when platelets were stimulated with complement, or integrin α2bβ3 signaling was triggered, ectosome formation was independent of calpain activity [43,45]. These data suggest that different agonists may have different effects on the cytoskeleton, resulting in changes to ectosome production. Additionally, these data suggest that while initial actin cytoskeletal rearrangement is involved in platelet-derived ectosome formation, the destabilization and reorganization of the cytoskeleton produces more ectosomes.

Upon platelet activation, cytosolic calcium levels increase, and phosphatidylserine (PS) residues are expressed on the outer membrane. The loss of PS asymmetry on the membrane coincides with platelet-derived ectosome formation [46]. The importance of PS in ectosome formation is elucidated by the decreased EV levels found in patients with Scott Syndrome who have defects in PS exposure [46,47,48]. However, not all ectosomes are positive for annexin V, which binds to PS.

### 2.3. Apoptotic Bodies

Apoptotic bodies are the largest extracellular vesicles and are often over 1000 nm in diameter (Figure 1). Apoptotic bodies are formed when a cell undergoes cell death, and the vesicles are formed from fracturing of the cell and its plasma membrane. This process is unregulated and simply a result of cell death. Many times, apoptotic bodies contain cellular organelles or fragments of organelles. In some pathologic conditions, platelet-derived apoptotic bodies have been found to contain mitochondria [9,35,49].

## 3. EV Characterization

The exact nomenclature of EVs is difficult to assign without direct visualization of EV production, thus the ISEV recommends that authors refer to EVs based on their physical characteristics. These characteristics can include size, density, biochemical composition, or cellular origin. To more specifically characterize EVs, the initial characterization to confirm the population should include 1: a quantitative measure, 2: a measure of EV abundance, 3: the presence of EV components associated with generic or EV subtypes, and 4: the presence of non-vesicular, co-isolated components [11]. Positive EV markers should include at least one transmembrane/lipid-bound protein, a cytosolic protein, and one negative protein marker [11]. Some generic EV markers include ALIX, tumor susceptibility gene 101 (TSG101), CD63, CD9, and CD81 for exosomes, ADP-ribosylation factor 6 (ARF6), vesicle-associated membrane protein 3 (VCAMP3), plasma membrane glycoproteins and annexin A1 for ectosomes and apoptotic bodies [11,50] (Figure 1). It is important to note that these may be altered depending on the cell of origin. For example CD9 is absent from EVs originating from natural killer cells, B-cells and some mesenchymal stem cells [51]. The recommended negative controls are albumin and apolipoproteins A1/2 and B. However, negative controls may sometime pose a problem because cell components from different compartments can end up in EVs and this could be increased in pathologic conditions [25,35,49]. Some specific EV markers are utilized to mark the cell of origin (Table 1).

There are many different techniques used to measure EVs, and each has its strengths and weaknesses. Some of the techniques used to image and measure EVs and their contents are flow cytometry, nanoparticle-tracking analysis (NTA), resistive pulse screen, atomic force microscopy, electron microscopy, super-resolution microscopy, mass spectrometry, raman spectroscopy, flourier-transform infrared spectroscopy, sulfo-phospho-vanilin assay, total protein analyses with a Bradford assay, enzyme-linked immunoassay (ELISA), fluorimetric assay, and global protein stain on SDS-PAGE. The numbers of EVs detected varies based on the isolation, storage, and quantification techniques and whether pre-requisites to EV definition are set (i.e., annexin V positive). Due to these differences, the numbers of total EVs, annexin V positive EVs, and platelet-derived EVs is varied across the literature. For example, Arauna et al. reported 5000 total EVs/µL plasma, 223 annexin V positive EVs/µL plasma, and 200 platelet-derived EVs/µL plasma [52]. While, Arraud et al. observed 11,500 EVs/µL plasma [7], Sabatier et al. observed 752 EVs/µL plasma and 625 platelet-derived EVs/µL plasma [6], and Terrisse et al. reported 7300 platelet-derived EVs/µL plasma [53]. Even with only the small selection of reports discussed here, there is enough variability in EV number and cell of origin to emphasize the importance of clearly presenting the isolation protocols and analysis techniques to appreciate the robustness of data and show the need for the inclusion of multiple techniques for EV characterization.

## 4. EV Clearance

Once EVs have been produced, they enter into circulation and are cleared within minutes to hours [54,84,85,86]. During this process, EVs are taken up by cells. This process occurs through fusion with the plasma membrane, receptor mediated uptake, or endocytosis (Figure 3). A target cell may take up the EVs to promote cell–cell communication or they may be taken up by scavenger cells. Macrophages appear to be an important scavenger, because depletion of macrophages in mice tipped the balance of EV secretion/clearance resulting in increased plasma EVs due to decreased EV clearance [84]. Macrophages will phagocytose annexin V positive EVs in circulation [87]. This clearance mechanism appears to be dependent in part on lactadherin binding to PS and the spleen. Mice missing either lactadherin or a spleen had increased platelet-derived EVs in the plasma [88]. Lactadherin on splenic macrophages recognizes the PS on the surface of EVs and mediates phagocytosis. Mice lacking lactadherin had increased plasma platelet-derived EVs and demonstrated an increased level of coagulation and thrombosis [89]. The shape of EVs is important for non-opsonized phagocytosis while the size affects antibody-enhanced phagocytosis by macrophages [90]. Interestingly, IgG coating of EVs enhanced phagocytosis independent of size, while IgM coating of EVs enhanced the phagocytosis of only smaller particles (less than 2 microns) (Figure 3), indicating a role for IgM in EV clearance by macrophages [90]. However, EV uptake is not specific to macrophages as EVs can be taken up by other cell types, such as endothelial cells, in a process that can also be dependent on anionic PS and lactadherin. The addition of neutralizing/blocking antibodies for annexin V and lactadherin greatly decreased EV uptake by human umbilical vein endothelial cells (HUVECs) [53]. The inhibition of PS and lactadherin did not completely diminish EV uptake, indicating that there is more to this process. For example, annexin V negative EVs cannot bind to lactadherin and would need to be cleared up by a different mechanism. One potential mechanism for uptake of EVs irrelevant of their annexin V status is via glycosylation/sialyation [91]. As EVs are cleared from circulation, a forward feeding process occurs where EV phagocytosis by macrophages induces apoptosis and EV production from said macrophage in a never-ending loop.

## 5. Production of Platelet-Derived EVs

### 5.1. Common Stimuli

Extracellular vesicles produced from platelets (PEVs) and megakaryocytes (MEVs) are a means of communicating with other cells in the body in ways that the platelet itself cannot communicate [92]. While EVs from these two cell types are very similar, there are differences. First, megakaryocytes constitutively secrete EVs, while platelets largely release them upon activation [37]. Platelets at rest, or in storage, can also produce EVs [43,55,93], indicating that time outside the body is enough of a stimulus to induce EV formation. Ponomareva et al. found that even 15 min under ex vivo conditions is enough for resting platelets to produce EVs [55] and the numbers increase greatly when platelets are kept in storage over several days [93]. These spontaneously released PEVs do not contain any organellar content and are circular and smooth, indicating that they are exosomes or ectosomes and not apoptotic bodies. Second, MEVs contain CD41, CD41b, PS, and full length filamin A, while PEVs contain CD41, CD42b, PS, LAMP1, and CD62P [37,47,56]. Filamin A is one of the major factors differentiating MEVs from PEVs. Third, EVs are produced along slender unbranched macropodia in megakaryocytes [37] while platelet EV production specifics depend on the platelet stimuli.

Any stimuli that can activate platelets will often result in the formation of PEVs and the quantity, structure, and size of the PEV released depend on the stimulus. These stimuli include shear stress, fibrinogen, collagen, thrombin, ADP, calcium ionophores, activated protein C, and epinephrine. It was shown that increased shear stress resulted in a higher number of PEVs, and this was markedly increased at 5 min of exposure to shear stress compared to 2 min [94]. Shear stress promotes PEV production through the binding of von Willebrand factor (VWF) to platelet glycoprotein 1b (GP1b) followed by calcium influx and the activation of platelet calpain [94,95,96]. Under experimental conditions, thrombin is often used for PEV production because it results in the release of a higher number of PEVs [35,55]. Ponomareva et al. showed that the increase in released EVs in response to thrombin, unlike other common platelet stimuli (arachidonic acid, ADP, collagen, and calcium ionophore), is due to their generation from the plasma membrane as well as from intracellular organelles, such as the open canalicular system [55]. In contrast, Miyazaki et al. found that PEV numbers were highest when they used high shear stress or fibrinogen as stimuli [94], while Sims et al. found that using calcium ionophore A23187 resulted in the most PEVs [48]. These disparate results could be partly explained by whether studies examined annexin V positive or negative PEVs or from the experimental conditions used. Ponomareva et al. saw that while the calcium ionophore A23187 stimulated the most PEV release, these PEVs were mostly annexin V negative [55]. Platelets stimulated with ADP or epinephrine showed very little PEV release and there was no prothrombinase activity associated with the vesicles [48,94,95]. In general, using a combination of stimuli, such as thrombin and collagen, results in the highest number of PEVs [37,48,94,97]. The release of PEVs in response to common stimuli is shown in Figure 4. The use of multiple stimuli perhaps mimics the vasculature in vivo and can explain why the activation and consumption of platelets during thrombosis results in the release of large quantities of PEVs [56].

With regards to size and shape, PEVs from resting platelets are variable in size (30–500 nM). Thrombin-stimulated PEVs are relatively small (50–100 nM diameter) and are spherical or elongated with a rough surface or thin offshoots [55]. Collagen-stimulated PEVs are in the middle size range (50–300 nM) and are circular and smooth [55]. Calcium ionophore-stimulated PEVs are more heterogenous and larger in size and have a rough membrane surface [55,97]. PEVs produced from these stimuli showed procoagulant activity [41,44].

The contents of PEVs also differ based on which stimuli were used to activate the platelets. In general, PEVs contain proteins, glycoproteins, lipids, RNA, transcription factors, sugars, and coagulation factors [48,98,99]. When stimulated with thrombin, or a similar stimulus, they also contain cellular organelles, such as mitochondria and their DNA [35]. PEVs can also contain factors that promote inflammation, such as complement, CD63 and ROS [37,100,101]. When stimulated with thrombin and collagen together or with calcium ionophore A23187, PEVs expressed more CD62 [37]. PEVs generated with both thrombin and collagen had glycoprotinIIb/IIIa complexes that bound to fibrinogen while the complement-induced PEVs did not express fibrinogen receptor function. Additionally, PEVs from thrombin (alone) stimulated platelets had increased levels of NADPH Oxidase 1 (NOX1) on their surface, causing increased superoxide generation [100]. These data are only a small selection demonstrating that differential stimulation of platelets results in PEVs that varies significantly.

### 5.2. Pathogenic Stimuli

While the production of PEVs can be triggered by common thrombogenic stimuli, their production can also be stimulated by pathogenic stimuli. Interestingly, activation with these different stimuli can result in PEVs expressing different contents. For example, stimulation of platelets with Dengue Virus results in PEVs that have a different EV marker expression. While stimulation with Dengue Virus or thrombin resulted in similar levels of CD41, heat shock protein 70 (HSP70), and CD63, and neither had CD81 expression, only the PEVs from thrombin-stimulated platelets had expression of CD9 [102]. Similar phenotypes were seen where LPS stimulation of platelets resulted in IL1β positive PEV production, while stimulation with thrombin resulted in interleukin 1 beta (IL1β) negative PEVs, and TNF receptor-associated factor 6 (TRAF6)-stimulated platelets had decreased PEV production overall [103]. As another example, platelets stimulated with NONOate or LPS, but not TNFα or thrombin, generate PEVs with more redox potential [104]. Staphylococcal superantigen-like protein 5 (SSL5) can also induce annexin V positive PEV formation [105]. Only a few pathogenic stimuli have been studied, but there is an indication that other pathogenic stimuli may also generate PEVs with different characteristics [83]. These data indicate that PEVs in different pathologic conditions are not equal and will account for different pathological responses. Where one treatment in a disease with increased LPS may help prevent PEV activity, the same treatment in a disease with increased SSL5 may have no effect or could even have a negative effect. For example, if patients are given adrenaline, their PEVs contain more dihydroethidium (DHE) fluorescence, indicating more ROS production [106]. Studies in future should focus on better characterization to understand the mechanistic activities of PEVs under different disease conditions and how different pathogenic stimuli are affecting PEV production and formulation.

## 6. Platelet-Derived Extracellular Vesicles under Disease Conditions

Platelet-derived extracellular vesicles (PEVs) interact with multiple cells, at various locations, and under several disease conditions. The functions of PEVs depend on where the PEVs are localizing. Macoux et al. observed that fluorescently labeled PEVs injected into mice mainly traveled to the spleen and lymph nodes, followed by the bone marrow, the liver, the lungs, and the kidney [85]. These data indicate that not only do PEVs have functions in the blood but they can also affect pathologies in these locations within the body. This is particularly important when considering the possibility of PEVs affecting coagulation outside the blood in pathologic conditions [107,108,109,110,111,112,113,114,115,116,117,118]. In particular, PEVs may play a large role in the coagulation of lymphatic fluid, which is normally a hypocoagulable fluid due to the paucity of phospholipids [107]. In the blood, PEVs will adhere to platelets, leukocytes (mainly neutrophils and monocytes), lymphocytes, and endothelial cells and can affect these cells [53,85,92,97,119,120,121]. PEVs have a pro-inflammatory and pro-thrombotic nature [1,53,92,119,120,122,123,124], such that PEV membranes have 50 to 100-fold higher procoagulant activity than calcium ionophore-activated platelets [125]. There are two main mechanisms by which PEVs are thought to affect coagulation (Figure 5). The first is through the presence of anionic phospholipids. The negative charge causes an interaction with cationic domains of clotting factors, providing a surface for the assembly of these factors and thus initiating thrombin generation. PEVs also contain clotting factors on their surface including factor Va, factor VIIIa, factor X, and lactadherin [125,126]. The second is through TF. TF on PEVs will form a complex with factor VII/VIIa to initiate the coagulation cascade. While the presence of TF in platelets is controversial [127,128,129,130,131], PEVs are often described as containing TF. The TF on PEVs could originate from the platelet or its uptake from TF-bearing EVs of other cellular origins [132]. Alternatively, PEVs can affect coagulation through the activation of other cells to induce a pro-thrombotic environment [53,57,58,103,105,120,121,133,134,135,136,137,138,139,140,141,142,143,144,145] (Figure 5). As an example, Terrisse et al. found that PEVs will adhere to endothelial cells dependent on lactadherin, PS, and αVβ3 integrin. This interaction triggered ROS production in endothelial cells which increased the amount of VWF expressed on the endothelial cell surface. Circulating platelets at this site would then bind to surface VWF via GP1b and p-selectin to start adhesion and clot formation [53]. In general, PEVs support thrombin generation and this correlates with increased fibrin clot formation [1,122]. The PEVs accelerate fibrin polymerization and support the formation of more compact clots that can resist fibrinolysis [122].

The extent to which PEVs can increase thrombosis depends on both their number and composition. PEVs from pathologic conditions, such as disseminated intravascular coagulation (DIC), thrombocytopenia, and systemic lupus, are larger in diameter [55]. These larger PEVs can possess mitochondria and are more prevalent in cardiovascular disease [49]. Additionally, these larger PEVs can process and present ovalbumin on major histocompatibility complex 1 (MHC-1) molecules through an active proteosome [85], indicating that these PEVs contribute to adaptive immunity which can then lead to a more pro-inflammatory and pro-thrombotic environment. Arauna et al. found that frail adults had higher concentrations of PEVs in the circulating plasma and that these PEVs had higher levels of PS and TF on their surface [52], while Grande et al. found that PEVs from obese women were more heterogenous in size and protein contents with an enrichment of thrombosis related proteins [146].

## 7. Platelet-Derived Extracellular Vesicles in Autoimmunity

Patients with autoimmune diseases are at higher risk of thrombotic complications. While many factors contribute to thrombosis in these patients, pro-inflammatory and pro-coagulant PEVs are often increased under these conditions, including Rheumatoid Arthritis (discussed below), Systemic Lupus Erythematosus (lupus) (discussed below), Sclerosis (discussed below), Antiphospholipid Syndrome [57,59], Diabetes Type 1 [6,53,147], Inflammatory Bowel Disease [148], Multiple Sclerosis [149], Raynaud’s Phenomenon [60] and Autoimmune Thyroiditis (both Hashimoto’s and Grave’s diseases) [150]. For the majority of patients, PEV levels decrease upon remission (discussed below). Many of these PEVs are positive for both IgG and IgM and socan form immune complexes and will have pro-coagulant activity. However, different disease states will favor one antibody over the other and will express different pro-coagulant proteins. For example, there are more IgG-positive PEVs in lupus and more IgM-positive PEVs in Multiple Sclerosis [149,151]. Understanding the composition of PEVs in these different disease states will improve understanding of their pathologic roles and therefore, elucidate potential therapeutic targets.

### 7.1. Rheumatoid Arthritis (RA)

Rheumatoid arthritis (RA) is a chronic, inflammatory autoimmune disease that causes problems within and outside of the joint. RA patients have increased inflammation with increased risk of atherosclerosis and cardiovascular disease and decreased life expectancy, which can result from acute myocardial infarction, stroke, and heart failure. In the plasma of RA patients, the blood thrombi are denser and contain fibers that are thinner and more tightly packed together than those in non-RA blood thrombi, making them less likely to fibrinolyse [61]. These blood thrombi can also contain EVs and PEVs [61]. In RA, levels of circulating PEVs are increased [57,61,133,152,153], and correlate with clinical manifestations including vascular manifestations, inflammatory parameters, and coagulation potential [57,61,134,152,153].

Interestingly, synovial fluid, which is normally non-clotting in nature, can develop thrombi during inflammatory episodes, such as in RA or arthropathies. This occurs through the leakage of inflammatory clotting factors into the synovial fluid [113,114], leading to coagulation and the need to anticoagulate the synovial fluid before analysis [112]. There is also leakage of nuclear cells into the synovial fluid, including fibroblasts and immune cells such as neutrophils, macrophages, and lymphocytes. The types of cells in the synovial fluid reflect the disease etiology, whether there is inflammatory arthropathy (sepsis, gout), or rheumatoid disease vs. an injury to the joint [113]. In RA, PEVs can leak into the synovial fluid, providing TF and a phospholipid surface on which the thrombin clot can form [57,134,152,154]. The PEVs that enter into the synovial fluid of RA patients contain more mitochondria and hyaluronan than the PEVs in osteoarthritis patients [35,155]. They also are positive for citrullinated peptides (cit-fibrinogen) which can lead them to form immune complexes [133]. In fact, the majority of immune complexes in the synovial fluid of RA patients have PEV [120] and are highly pro-inflammatory.

These PEV-immune complexes can activate immune cells such as neutrophils and monocytes. PEV-immune complexes have been found to stimulate neutrophils resulting in them producing leukotriene [120], and stimulate monocytes inducing them to produce chemokine receptor 1 (CXCR1), CD36, and interleukin 1 (IL-1) [133]. However, PEVs can also activate other cells without being in an immune complex. PEVs in RA can modulate inflammation, endothelial cell activation, fibroblast activation, and coagulation [57]. PEVs can directly stimulate monocytes causing the release of IL-1β, IL-6, IL-8, IL-10, and TNFα [133]. PEVs also activate fibroblast-like synoviates in the synovial fluid through IL-1, platelet factor 4 (CXCL4) and platelet basic protein (CXCL7) on their surface [134,136]. IL-1 on the PEVs stimulates the synovial fibroblasts to produce IL-8 [134]. CXCL4 and 7 trigger nuclear factor kappa-B (NFκB) activation in the fibroblast-like synoviates, promoting their migration, which then results in joint erosion [136]. In a feedback loop, fibroblast-like synoviates can also induce PEV production through glutathione peroxidase (GPXI) and FC receptor gamma (FCRγ) chain signaling, and without GPXI, mice had decreased inflammatory arthritis [134].

While clotting in RA occurs in both the blood and synovial fluid, PEVs also enter the lymph of RA patients [156]. Interestingly, the EVs in the lymph of RA patients are mostly of platelet origin and not of erythrocyte or leukocyte origin, indicating a role specifically for PEV. However, the PEVs in the lymph of RA patients lacked mitochondria and were unable to promote coagulation in the lymphatic fluid, indicating that while they promote coagulation in other locations, they may serve more of an immune-regulating/activating role in the lymph.

### 7.2. Systemic Lupus Erythematosus (Lupus)

Blood thrombi are seen with increased frequency in individuals with lupus. One reason this may happen is that these patients have anti-phospholipid antibodies (APS). However, lupus is a complex disease and the presence of APS antibodies is not the only factor affecting coagulation [80]. Another potential factor leading to increased thrombi formation in lupus patients is platelet activity and the increased prevalence of PEVs. In lupus, the platelets are smaller and more activated [157]. The prevalence of smaller platelets is most likely a consequence of larger platelets being consumed preferentially, resulting in more PEVs being produced (presumably from the consumed larger platelets). Interestingly, decreased platelet size is associated with increased anti-phospholipid syndrome, and thrombotic events in lupus [157]. As such, PEVs in lupus likely are promoting anti-phospholipid syndrome and thrombosis. PEVs are elevated in the circulation in lupus by 2 to 10-fold compared to healthy controls [10,49,60,158]. In patients with higher levels of circulating TNFα, there are also increased PEV numbers [62] indicating that TNFα may be stimulating PEV production in lupus. On the other hand, researchers have also found the opposite and reported that PEVs decrease in lupus [72]. One potential reason for this controversy could be that the PEVs are being consumed, leading to disease activity. Alternatively, in lupus PEVs form immune complexes with both IgG and IgM [57,151,159,160,161] and the PEV-immune complexes are likely being deposited in places such as the kidney glomeruli, leading to loss of circulating PEVs and perhaps pathologies like lupus nephritis. However, some studies have reported that patients with lupus nephropathy have increased circulating PEV [63], which may reflect a discrepancy in methods used to analyze the PEVs.

In lupus, the majority of circulating IgG immune complexes are found on plasma EVs [159]. While IgG-PEVs correlate with disease activity, IgM-PEVs do not, indicating that the type of immune complex formed with the PEVs impacts its pathogenicity [159]. Some other prothrombotic factors clinically associated with IgG-PEV complexes are dsDNA antibodies, anti-histone antibodies, total IgG, decreased leukocyte counts, and the Systemic Lupus Erythematosus Disease Activity Index (SLEDAI) score [151,161]. IgG-PEV complexes also activate monocytes in vitro and lead to the increased production of CD69, CD64, IL1β, TNFα, and IFNα from the monocytes [159]. While most of the PEVs are likely to be associated with immune complexes, researchers have also observed that PEV numbers themselves correlate with disease duration, thrombin generation, and cardiovascular disease in lupus [10,62]. Though PEV-immune complexes correlate with active disease, PEVs are increased even in the absence of any active disease [10,159]. APS antibodies in lupus were not associated with PEV levels [159], and patients with no APS also have increased total EVs that associate positively with thrombin generation [80].

In general, PEVs from lupus are larger in diameter than those in healthy individuals [55]. A proportion of these large PEVs (larger than 700 nm) contain mitochondria [49]. Patients with increased numbers of mt-positive PEVs also had worse disease outcomes (dsDNA antibodies, pro-inflammatory cytokines, and lupus nephritis). Smaller PEVs in lupus were determined to have decreased mitochondrial proteins but increased glycolytic and apoptotic proteins, including annexin V, CD40 ligand, and galectin-3 binding protein G3BP [78,158,160]. Increased amounts of glycolytic and apoptotic proteins in PEVs indicated an increased SLEDAI score [78]. These PEVs also had increased levels of complement proteins [10,151,158,160].

### 7.3. Vasculitis and Systemic Sclerosis

Vasculitis is a complication of many autoimmune diseases, including scleroderma, or systemic sclerosis (SS). During vasculitis or SS, patients have a procoagulant state and have an elevated risk of venous thromboembolism [162]. Levels of PEVs are increased during vasculitis occurring at any age and often correlate with disease symptoms. In sclerosis, Guiducci et al. found that there were increased levels of PEVs in both the limited and systemic forms of the disease [163]. Not only are PEVs elevated in SS, but EVs originating from endothelial cells, monocytes, and T cells are also increased. However, PEVs are the main source of plasma-derived EVs in SS, where 67% of EVs are platelet-derived and endothelial cell-derived EVs, constituting merely 9%, are the second largest source [163]. The elevation of PEVs in SS is greater than that of RA [73]. PEV levels also correlated with disease activity, including the Rodnan skin thickness score, and the Birmingham vasculitis activity score (BVAS) [64,163], and decreased at remission [73]. Patients experiencing interstitial pneumonia, or nephropathy had increased PEVs in circulation [63,164]. While the correlation of specifically PEV levels has not yet been examined for correlation with vasculopathy as a symptom, total EV levels in SS are associated with increased vasculopathy [74].

Patients with SS can have anti-neutrophil cytoplasmic antibodies (ANCAs) [165] which by themselves can lead to ANCA-associated vasculitis. In patients who have ANCAs, platelets are activated via the thrombin-protease activated receptor (PAR) pathway and produce PEVs [162]. During vasculitis, whether ANCA positive or not, PEVs are increased and are associated with active disease, inflammation (immune cell counts, erythrocyte sedimentation rate (ESR), C-reactive protein (CRP)), and renal damage [63,64,73,74,147]. PEV levels were found to be increased in chronic vasculitis with even greater numbers in acute vasculitis. One group did not see an increase in PEV levels in patients with Coronary Artery Disease, a vasculopathy independent of SS [166,167], indicating a role for PEVs in SS-associated vasculitis. Leleu et al. did not see increases in endothelial cell-derived EVs while they observed significant changes in PEVs [74]. Intriguingly, PEVs in vasculitis seemed to be increased if they expressed CD42a but not if they expressed P-selectin [64], indicating that the characteristics of EVs may align with the disease state.

PEVs in SS and vasculitis express many pro-inflammatory and pro-coagulant proteins, including TF, IgG, sialic acid-binding immunoglobulin-type lectin 11 (siglec-11), tumor necrosis factor receptor superfamily member 19L (RELT), Nectin 1 and 4, TFN RII, IL1-R4, junctional adhesion molecule B (JAM-B), Cadherin-11, 13 and P-cadherin, glypican 5, thrombospondin 2 and 5, CD23, granulocyte colony-stimulating factor (G-CSF), angiotensinogen, UL16-binding protein 1 and 2 (ULBP-1 and 2), interstitial collagenase (MMP-1), CD229, CD84, CD58, and IL-12p40 [147,151,160,164]. As such it can be inferred that they probably play roles in chemotaxis and adhesion of immune cells, cell growth, apoptosis, and coagulation. PEVs expressing the protein high mobility group box 1 protein (HMGB1) caused neutrophil activation and neutrophil extracellular trap (NET) formation (NETosis) through this protein [166,167].

## 8. Platelet-Derived Extracellular Vesicles in Allergies

There are indications that activation of the coagulation system occurs during allergic responses [168,169,170]. The role of the coagulation system would depend on the type of allergy, but one study indicated that thrombin plays an important role in the induction and regulation of mucin production in the upper airway [169]. It has been found that people with seasonal allergic rhinitis and asthma have increased plasma levels of platelet activation markers, including PEVs [65,171]. No significant changes in EVs from endothelial cells were observed [65], indicating a role for PEVs. Additionally, PEVs were increased in patients with atopic dermatitis and decreased as symptoms disappeared [172]. This was in comparison to both healthy and non-atopic urticaria patients, indicating a role of immune-mediated PEV production. PEV numbers also correlated with the scoring atopic dermatitis (SCORAD) index, indicating a potential role of PEVs in the disease state. While there are many unknowns in the roles of PEVs in allergic responses, there are a few possible mechanisms that it would be beneficial to investigate. First, PEVs contain and can deliver CD154 [173] and platelets promote allergic asthma through expression of CD154 [174]. Next, PEVs can be positive for Platelet-factor 4 (PF4) [175] and PF4 increases in people with seasonal allergies when they are experiencing symptoms [171]. Further investigation of the role of PEVs in allergies could provide new insight into the pathologic mechanisms of different allergic responses.

## 9. Platelet-Derived Extracellular Vesicles in Infectious Diseases

Many different infections or infectious diseases have pathologies involved with altered coagulopathy and thrombosis. These thrombo-inflammatory disease manifestations can be caused by a wide variety of etiologies, from bacteria to viruses to parasites and their resulting systemic inflammation. Infection itself can lead to thrombosis, or infection can lead to the release of pro-inflammatory components that then promote thrombosis. In many infectious diseases, there is a combination of both. Many of these stimuli induce PEV formation, and these PEVs can then enhance thrombosis in inflammatory and infectious conditions. It is important to consider that the pathogen itself can affect PEVs. Not only can they directly stimulate their release, but components of the pathogen can be present in the EVs [66,176,177,178,179]. In some cases, the bacteria or parasites can produce their own EVs, which could impact EV detection and characterization [180,181]. Below, we present a selection of diseases in which PEVs have been demonstrated to affect patient coagulation states.

### 9.1. Sepsis

Sepsis is a disease in which the body responds to an infection and mounts a massive immune response which spirals out of control. Mortality in sepsis is high globally, and one cause of death is disseminated intravascular coagulation (DIC). Thrombocytopenia in sepsis is correlated with a poor clinical outcome and also correlates with high numbers of PEVs [182]. In the plasma of septic patients, multiple groups have found high PEV numbers [79,104,182,183,184,185] that correlate with poor clinical outcomes. PEVs correlated positively with septic shock [75,79], DIC [75,79,182], renal dysfunction [185], longer vasoactive support [79], and longer ventilation [79]. This was not true for EVs originating from endothelial cells [75]. PEVs specifically were not found to correlate with mortality, but CD31 positive EVs, which include a majority of PEVs, do correlate positively with mortality [75]. Curiously, another group found that PEV levels decreased with DIC and mortality [183]. This discrepancy is likely due to PEVs associating with thrombi [122] in the patients so that they were not available in the aspirated blood. The increase in PEVs in sepsis is associated with acute disease [79,175]. However, in patients who suffered DIC, PEVs remained elevated for up to 21 days after DIC treatment [182]. The initial infection causing sepsis affects PEV numbers and contents. For example, patients with fungal sepsis caused by *Candida albicans* had elevated levels of PEVs compared to those with non-fungal sepsis [186]. Furthermore, patients with sepsis caused by *Neisseria meningitidis*, which causes massive coagulation, had increased levels of TF-positive EVs [83]. All these data indicate that PEVs are an important factor in the progression of sepsis pathology, and due to the diverse nature of sepsis, the PEVs and their downstream effects are not always the same for every patient.

The cytokine storm in sepsis can mediate platelet activation and PEV release. Some of the cytokines indicated to be involved in the release of PEV in sepsis are IL-6 [182] and IL-8 [81]. Pathogenic components can also activate platelets to release PEV [103,105,187,188]. The pathogenic agonists appear to have a greater propensity to stimulate PEV formation in sepsis. For example, it has been shown that LPS stimulation results in IL-1β rich PEV, while stimulation with a TRAF6 interacting peptide-antennapedia chimera does not cause the release of PEV despite causing platelet activation [103].

PEVs in sepsis lead to a prothrombotic environment not only due to an increase in numbers, but also through their contents, including PS [184], and their ability to activate other cells. Wang et al. found that PEVs can cause neutrophil cell death, potentially via NETosis, through Rac1 [135]. Inhibition of Rac1 resulted in a reversed effect on thrombin generation in sepsis. Monocyte activation can also be mediated by PEVs in sepsis. Platelets stimulated with the Staphylococcal superantigen-like protein 5 (SSL5) produced PEVs that bound monocytes with a preference for intermediate monocytes, then classical monocytes, and last non-classical monocytes. The interaction of PEV and monocyte-induced cytokine production from the monocytes, including IL1β, TNFα, monocyte chemoattractant protein-1 (MCP-1), and matrix metalloproteinase 9 (MMP-9) [105]. These PEVs also induced monocyte migration that was greater than SSL5 was able to induce. Additionally, PEVs were able to recruit monocytes through the interaction of PS and p-selectin [121]. This interaction resulted in transferring GP1bα to the monocyte from the PEV. As such, there was greater tethering and rolling of monocytes to the VWF exposed on endothelial cell surfaces. Normally, VWF is not exposed on the surface of endothelial cells, but PEVs in sepsis can lead to this exposure [137]. PEVs can also activate endothelial cells through the IL-1 receptor, resulting in elevated vascular adhesion molecule 1 (VCAM-1) expression [103]. PEVs can also be internalized by endothelial cells [138], and the NADPH on their surface can lead to ROS production and endothelial cell apoptosis [104,139]. Just like sepsis, this is more complicated than meets the eye. In a likely attempt by the body to control and cool down the system, PEVs in sepsis are also enriched in miR-223 which reduces intracellular adhesion molecule 1 (ICAM-1)-dependent vascular inflammation in sepsis. This leads to decreased adhesion of peripheral blood mononuclear cells (PBMCs) to the endothelial cell surface [138] and may decrease thrombo-inflammation.

### 9.2. Human Immunodeficiency Virus (HIV)

Patients with HIV infection have a hypercoagulable state that increases their risk of thrombotic complications even without active viral replication [140,189,190]. In these patients, an increased viral load also leads to increased platelet activation [191]. One way in which this occurs is through the protein transactivator of transcription (Tat). HIV Tat1 directly activates platelets through chemokine receptor 3 (CCR3) and β3-integrin, resulting in PEV formation [191]. Correspondingly, patients with HIV have increased numbers of PEVs in their blood [9,58,76,192], though the numbers of PEVs detected are again variable in different reports. Hijmans et al. found 40 PEVs per µL plasma in healthy individuals and 140 PEVs per µL plasma in HIV patients [76], while Falasca et al. found 897 annexin V negative and 112 annexin V positive PEVs per µL plasma in healthy individuals and 1419 annexin V negative and 215 annexin V positive PEVs per µL plasma in HIV patients [192]. The PEVs in HIV patients contain protein components that are involved in thrombus formation and inhibition of the immune response. Falasca et al. identified that some proteins involved in the thrombotic response of PEVs are apolipoprotein E (APOE), Beta-2-glycoprotein 1 (APOH), complement component 3 (C3), complement component 5 (C5), complement factor H (CFH), complement factor H-related protein 1 (CFHR1), factor 2 (F2), fibrinogen alpha (FGA), fibrinogen beta (FGB), fibrinogen gamma (FGG), histidine-rich glycoprotein (HRG), kininogen-1 (KNG1), plasminogen (PLG), and antithrombin-III (SERPINC1) [192]. It has also been found that PEVs in HIV have higher numbers that are positive for dihydroethidium (DHE), indicating that they contain ROS [106]. Unexpectedly, even though there is increased ROS, these PEVs actually have lower mitochondria [9], suggesting sources other than mitochondria induce ROS production in PEVs in HIV.

PEVs, and even MEVs, contain HIV viral receptor C-X-C chemokine receptor type 4 (CXCR4) [141]. The components on the surface of PEVs, including CXCR4, can be transferred to cells that are devoid of these components. The transfer of CXCR4 from PEVs to cells provides a significant pathological trigger where PEVs could then potentiate the spread of the virus through the transfer of the viral receptor to cells naturally devoid of these receptors [141]. Other than spreading the virus, PEVs in HIV correlate with decreased levels of CD4-positive T cells [58] and can cause endothelial dysfunction [76]. These data indicate that PEVs play a role in the pathogenicity in HIV patients through their participation in cellular interaction and coagulation.

### 9.3. COVID-19

COVID-19 disease is characterized by thrombo-inflammation. Microthrombi form in the lungs of patients and larger thrombi form in circulation, leading to thrombotic events. Platelet activity has been found to be increased in COVID-19 patients, and PEVs are released as a result [77,193]. While exact numbers of PEVs detected differ, PEV levels are increased in the circulation of COVID-19 patients [67,68,77,142,143,194,195,196,197,198]. Cappellano et al. visualized 1472 PEVs/µL plasma, while Maugeri et al. observed 2.5 × 10^6^ PEVs/µL plasma, and Zaid et al. detected 2 × 10^8^ annexin V negative PEVs/µL plasma and 9 × 10^8^ annexin V positive PEVs/µL plasma [194,195,196]. Additionally, PEVs in COVID-19 plasma are larger than those found in healthy controls [143], indicating that there may be a higher ratio of ectosomes and apoptotic bodies in circulation.

The enhanced prevalence of PEVs correlates with COVID-19 disease severity. While asymptomatic patients had increased levels of PEVs compared to healthy controls, PEV levels rose dramatically in individuals with active disease [142,194,198]. If a patient was hospitalized, the PEV levels were the highest in those admitted to the ICU [198]. PEV levels correlated with longer hospitalization, intubation, total leukocyte counts, neutrophil counts, CRP expression, lactate dehydrogenase (LDH) levels, and thrombotic events [68,142,196,197]. Conversely, Zaid et al. and Guervilly et al. observed decreased levels of PEVs in the circulation of patients with severe COVID-19 and thrombosis [82,196]. This discrepancy is probably due to the consumption of PEVs that are then incorporated into thrombi [122]. This hypothesis is further supported by the finding that there is a negative correlation between PEV numbers and d-dimer levels, suggesting PEVs are being consumed in DIC [68]. PEVs that are HMGB1 positive are increased in COVID-19 and correlate positively with CRP levels, LDH expression, d-dimer, acute respiratory distress syndrome (ARDS), and patient WHO score. These PEVs correlate negatively with PaO2/FiO2 in COVID-19 [195]. Additionally, cancer is a co-morbidity often associated with COVID-19, and PEV levels are high in patients with both cancer and COVID-19. However, only PEVs expressing platelet activation markers, such as CD146, showed further increase upon metastasis. These activated PEVs correlated positively with the increased levels of CRP and d-dimer in metastatic patients [77]. Total EVs in COVID-19 are enriched for coagulation, inflammation, and the immune response, signifying their ability to affect these clinical outcomes [199].

As the COVID-19 pandemic progressed, it became apparent that many individuals would suffer from symptoms weeks to months after they became sera-negative for the virus. This phenomenon has been termed post-COVID. As of yet, there is no identifying reason as to why one person will get post-COVID, and another will not. In studies examining PEV levels after acute disease, two groups found contrasting results. Campello et al. observed that PEVs remained elevated in the plasma of COVID-19 patients for at least 30 days post-hospital discharge [67]. They also observed that EVs originating from endothelial cells decreased after recovery, indicating a potential role for PEVs, specifically, in post-COVID symptoms. On one hand, numbers of PEVs were positively associated with thrombosis and with persistent symptoms. On the other hand, Abdelmaksoud et al. observed a decrease in PEV levels four weeks after patients were sera-negative for SARS-CoV-2 [197]. In this study, the authors did not observe post-COVID symptoms, but they did see that the levels of PEVs correlated positively with CRP, LDH, and serum ferritin. These different observations for PEVs in post-COVID patients could be due to the non-inclusion of patients without post-COVID symptoms. There is a strong likelihood that elevated PEVs are associated with persistent symptoms as both groups observed that increased PEV numbers correlated with symptoms after initial COVID-19 recovery.

In COVID-19, there are many stimuli in circulation that can lead to PEV formation. A few predicted PEV-stimuli in COVID-19 are IL-1, IL-6, TGFβ, and the virus itself [199,200]. Whether SARS-CoV-2 interacts with platelets in vivo is not fully elucidated and has only been seen in a few patients [200,201]. Studies indicated that while infection may be possible, it is unlikely that platelets support viral replication [202]. In vitro experiments have been performed to determine the effect the virus has directly on platelets and if it induces PEV formation. It is important to keep in mind that the viral doses used in vitro may be substantially higher than those found interacting with platelets in vivo; it is also important to consider that factors in the bloodstream released from other cells the virus comes into contact with, can also activate platelets. In vitro, SARS-CoV-2 interaction with platelets activates them and results in cell death and PEV release [200]. This is both dependent on and independent of angiotensin-converting enzyme 2 (ACE2). Rarely has ACE2 RNA or protein been detected on the platelet surface [193,200,203], but there are other SARS-CoV-2 receptors on the platelet surface. CD147, a novel receptor for SARS-CoV-2 [204] has increased expression on platelets in COVID-19 infection [202]. Activation of the CD147 receptor on platelets resulted in increased release of PEVs [195]. However, the interaction of platelets and the virus is probably dependent on the multiplicity of infection (MOI). Koupenova et al. saw large effects as they used an MOI of 10, whereas Shen et al. saw fewer effects when they used an MOI of 1 [200,202].

Functionally, annexin V positive PEVs were observed to adhere to PBMCs for weeks after initial diagnosis [142]. The number of PEVs adhering to the PBMC correlated with increasing disease severity as measured by the WHO ordinal scale. PEVs bound to CD4^+^ T cells, CD8 T cells, CD19 positive B cells, and a fourth group of PBMCs comprised mainly of dendritic cells and monocytes. The PEVs preferentially bound to proliferating T cells with an increased propensity for effector T cells compared to memory T cells. Other than binding to immune cells, PEVs from COVID-19 plasma triggered endothelial cell activation and death [143]. Both p-selectin and VCAM-1 expression in endothelial cells increased via the annexin V on PEVs. This activation of the endothelial cells then resulted in activation of neutrophils, increasing their adherence and NETosis [143], indicating that PEVs in COVID-19 not only adhere to PBMCs, but they can also affect endothelial cells and modify the vasculature in a procoagulant way. With these activities, PEVs, and not platelets, have been proposed as a potential biomarker for COVID-19 disease [194].

### 9.4. Dengue

Dengue virus is another virus that upon infection of humans, results in DIC [205]. While only a small population of infected individuals experience DIC, those who do are classified as having dengue hemorrhagic fever. Infection leads to platelet activation and PEV production [144], which further results in thrombocytopenia. In patients infected with dengue who had thrombocytopenia and no bleeding phenotype, higher levels of PEVs are observed [69]. Those who bled had low levels of both platelets and PEVs, while those who did not have thrombocytopenia did not have high numbers of PEVs [66,69]. While PEV levels are lower in individuals who experience bleeding, this is most likely due to PEVs being associated with thrombi found in the bloodstream [122] such that they cannot be visualized ex vivo. Additionally, PEVs were found to play a role in dengue virus lethality [102].

PEVs are produced in dengue through the activation of platelets and the platelets can be activated through inflammatory factors or the virus itself. Dengue virus can activate platelets through multiple mechanisms, including through activation of the platelet c-type lectin-like receptor 2 (CLEC2), which is a receptor for the snake venom aggretin [102], and through the production of mitochondrial ROS [144]. When the CLEC2 receptor is stimulated with dengue virus, the PEVs produced can activate CLEC5A and Toll-like receptor 2 (TLR2) on neutrophils and macrophages, resulting in their activation. PEVs can also cause NETosis [102]. It was found that when the CLEC5A and TLR2 receptors are inhibited, dengue lethality was decreased from 70% to 10%, indicating the potential of PEVs in dengue-related mortality through this particular mechanism. Whether through a direct mechanism or an indirect one, upregulation of mitochondrial ROS in dengue infection led to the assembly of the nucleotide-binding domain leucine-rich repeat-containing protein (NLRP3) inflammasomes through caspase 1, which resulted in increased PEV production [144]. The PEVs produced were rich in IL-1β. The IL-1β on PEVs led to enhanced vascular permeability, as evidenced by leakage through an endothelial cell layer. While the PEVs in dengue had high levels of IL-1β, they were also positive for HSP70 and CD63. Interestingly, CD9 was absent in PEVs from dengue-stimulated platelets, while it was present in PEVs stimulated with thrombin [102]. These data indicate that the PEVs produced in infection with dengue virus are different from those produced without viral infection.

### 9.5. Malaria

Coagulation and DIC are a consequence of malaria. DIC is associated with more severe disease and cerebral malaria, and as such, high mortality [206]. In malaria infection, both in people (*Plasmodium falciparum* and *vivax*) and in mice (*Plasmodium berghei*) there are higher numbers of circulating EVs and PEVs [70,71,207]. PEVs in malaria positively correlate with clinical symptoms, including fever, cerebral malaria, coma depth, thrombocytopenia, and length of symptoms [70,71]. PEV levels did not correlate with severe anemia or uncomplicated malaria. Mfonkeu et al. observed levels of annexin V positive PEVs increased from 50 in healthy individuals to 150 PEVs per µL of plasma in malaria patients presenting with cerebral malaria [70]. In mice infected with *P. berghei*, mortality due to cerebral malaria was decreased by the addition of a caspase inhibitor, which reversed the inverse correlation between platelets and PEVs due to inhibition of the caspase-dependent disintegration of the platelet and the resulting PEV production [8]. Interestingly, levels of PEVs decreased with increasing numbers of previous malaria episodes [71], and were associated with acute disease, as PEV numbers decreased upon hospital discharge [70].

PEVs in malaria affect coagulation by interacting with and activating other cell types. First, PEVs can bind to and be taken up by parasitized red blood cells (pRBC) [145,207]. In doing so, the PEV transfers both cholesterol and platelet antigens to the surface of the pRBC. While the cholesterol serves as a necessary resource for pathogen replication [207], the platelet antigens facilitate cytoadherence to brain endothelial cells [145], promoting cerebral malaria. PEVs had minimal binding to uninfected RBC. Second, the PEVs can also directly interact with brain endothelial cells. This interaction leads to the acquisition of platelet endothelial cell adhesion molecule 1 (PECAM-1) and GPIV by the endothelial cells from the PEVs, providing a pro-coagulant environment [145]. Third, EVs, including PEVs, can be taken up by and activate human spleen fibroblasts, inducing ICAM-1 expression [208]. While these are three ways in which PEVs can lead to coagulation and adverse effects in malaria, there could be other mechanisms yet to be discovered.

## 10. Effects of Therapeutics on PEVs

An interesting developing concept which has been not explored much is that different treatment regimes can alter the production and characteristics of PEVs. One study found that low molecular-weight heparin (LMWH) led to a decrease in the procoagulant activity of PEVs in patients with deep vein thrombosis. However, if a vitamin K antagonist was used prior to the treatment with LMWH, the PEVs showed an increase in procoagulant activity [209]. On other hand, different therapies targeting the same pathway or receptor can also have different effects on PEVs. Two anti-platelet therapies antagonizing the P2Y12 ADP receptor, while resulting in similar effects on platelet aggregation, showed diverse effects on PEV production [210]. Ticagrelor treatment resulted in decreased platelet activity and PEV formation, while Prasugrel resulted in decreased platelet activity but had no significant effect on PEV production. These data are examples of how the choice of therapies, and even the order in which they are given, can affect PEVs and their resulting impact on disease states.

Additionally, treatments used in the prevention of disease can also alter the production of PEVs. For example, vaccine-induced thrombotic thrombocytopenia (VITT) can result from vaccination against SARS-CoV-2 [211]. In this condition, individuals have PF4-positive immune complexes [212] and these complexes have been found to induce platelet activation and PEV formation [213]. Therefore, knowledge of how vaccination affects PEVs and their potential to induce thrombosis is important in order to maintain an individual’s health.

## 11. A Feedback Loop of Platelet-Derived Extracellular Vesicles Activating Parent Cells

PEVs contribute to thrombosis through direct contribution with PS and TF, and indirectly through activation of other cells and creation of a pro-thrombotic environment. Interestingly, PEVs have a feedback loop where they can also act on platelets and their precursor, megakaryocytes. In this way, PEVs affect their parent cells and this loop can result in an increased number of pro-thrombotic PEVs. EVs derived from both platelets and megakaryocytes can enter the bone marrow, and promote hematopoietic stem/progenitor cell (HPSC) differentiation into mature megakaryocytes [96] (Figure 6). The addition of cycloheximide inhibited this phenomenon, suggesting that properly functioning protein synthesis is necessary for this event to occur [214]. PEV-stimulated megakaryocyte maturation is independent of thrombopoietin (TPO) (Figure 6) as direct injection of PEVs into irradiated mice resulted in increased levels of megakaryocytes and platelets without affecting TPO levels [214]. This is substantiated by the finding that ex vivo treatment of bone marrow from mice lacking the TPO receptor with wild-type PEVs, restores megakaryocyte differentiation [215]. In PEV-stimulated megakaryocyte differentiation, the megakaryocytes were generally larger and more polyploid [214] (Figure 6). The fragmentation of the megakaryocyte cytoplasm that occurs with ploidization, results in an increased platelet count as long as the cellular organelles (mitochondria, granules, and demarcation membrane system) are developed during megakaryocyte maturation [216]. PEVs contain miRNA1915-3p, which suppresses the expression of Rho GTPase family member B (RHOB) [214]. While RhoA plays a key role in the regulation of actin polymerization and megakaryopoiesis [217], RhoB plays a role in microtubule/myosin contraction to promote actin assembly on endosomes which in turn regulates protein trafficking and cell survival. As such, it is not surprising that inhibition of RhoB results in increased megakaryopoiesis [214]. Since PEV-stimulated megakaryocyte differentiation results in larger and more polyploid megakaryocytes, it is safe to assume that there will be a resulting increase in platelet count, and Qu et al. found that PEVs do indeed result in increased platelet-like particle formation in megakaryocytes in vitro [214] (Figure 6).

Not only can PEVs have an effect on the megakaryocyte, but they can also activate platelets directly. One way in which this occurs is through NADPH oxidase 1 (Nox1) on the PEV surface. The Nox1 on PEVs enhances fibrinogen binding, annexin V exposure, and p-selectin exposure in platelets [100]. Therefore, it is likely that the Nox1 on PEVs also results in increased PEV production from these activated platelets. However, as PEVs contain many molecules that activate platelets, it is likely that there are a multitude of ways in which PEVs can activate platelets through their receptors or from PEV uptake. In pathologic conditions, the number of PEVs in the bone marrow is increased [215], suggesting that this process of PEV affecting megakaryocytes and platelets is also increased in pathologic conditions, creating an endless loop where PEVs result in more EV production from platelets and other cells [87]. Since there are more PEVs in pathologic conditions, this cycle will continue to spin out of control, resulting in a worsened state for patients.

## 12. Future Directions

Understanding the role of PEVs in health and disease is important to appreciate their contribution to pathologic conditions. In particular, one role PEVs are known for is their ability to promote coagulation and a pro-thrombotic state [1,53,92,119,120,122,123,124,125]. The production of PEVs can be induced by classic platelet stimuli that originate from the host or from pathogens. Since different stimuli result in different numbers and phenotypic expression/contents of PEVs, it is important to know how PEVs are being generated in different disease states as well as to know how different agonists and therapies affect PEVs. Intriguingly, coagulation can also occur in fluid other than that of blood, including the bone marrow [108], the CNS [109,110,111], the lymphatic system [107,115,116,117,118], and the synovial fluid [112,113,114,156] in pathologic conditions. This occurs due to leakage of clotting factors and PEVs into these fluids allowing for coagulation, but very little has been studied about the roles of PEVs in coagulation outside the blood. This is important because PEVs can reach sites in the body that the platelet cannot [218]. Thus, a detailed understanding of PEVs in a pathologic setting and how the therapies used to treat said disease affect PEVs, is necessary in order to provide appropriate treatment to patients. Conversely, because of their role in thrombosis, PEVs have been considered as a therapeutic potential in wound-healing hemorrhage [219,220]. We have detailed the role of PEVs in coagulation in multiple immune-mediated diseases, but PEVs have roles in other disease states as well, including ones that have yet to be examined such as Ebola and Babesiosis which both have pathogenic effects on the coagulation system. In pathologic conditions, knowledge of what stimuli are used to produce the PEVs, and their composition, is just as important as understanding their pathologic role in disease.

## Figures and Tables

**Figure 1 ijms-23-07837-f001:**
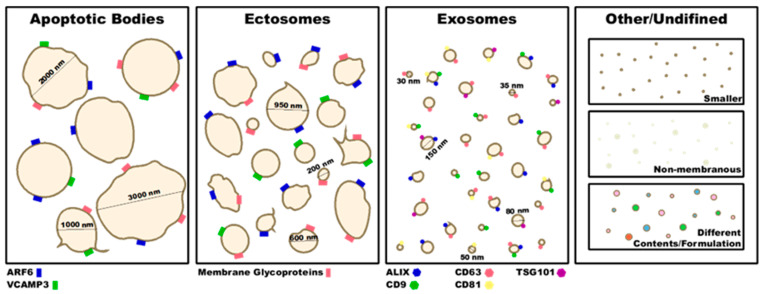
Subtypes of Extracellular Vesicles. There are three main subtypes of extracellular vesicles (EVs) that have been characterized to date, apoptotic bodies, ectosomes (microparticles/microvesicles), and exosomes. Apoptotic bodies are large (greater than 1000 nm in diameter) and heterogenous in size and shape due to them originating from plasma membrane blebs. Ectosomes, are also heterogenous in size and shape and can range from 150–1000 nm in diameter. Exosomes are more homogenous in size and shape, being 30–150 nm in diameter and mostly round, as they are produced via a regulated process., Apoptotic bodies and ectosomes originate from the plasma membrane and can express specific markers ARF6, VCAMP3, and membrane glycoproteins. Exosomes can be identified by the markers ALIX, CD9, CD63, CD81, and TSG101. While these three EV types have been characterized, emerging technologies are revealing new EV populations that have yet to be characterized. These include EV populations that are smaller, are non-membranous, and/or have different contents and mechanisms of production.

**Figure 2 ijms-23-07837-f002:**
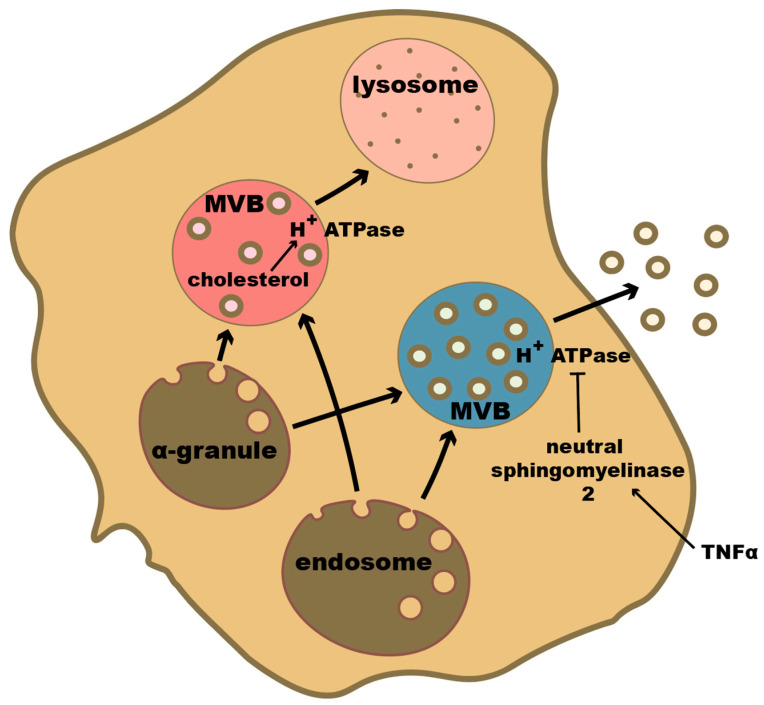
Effect of pH on Exosome Biogenesis. When the endosome, or the α-granule in platelets, undergoes invagination of its membrane, a multivesicular body (MVB) is formed. An acidic pH in the MVB destines the created exosomes for degradation via the lysosome, while a more basic pH destines the exosomes for secretion. H^+^ ATPase controls the acidification of the MVB. Cholesterol in the MVB activates H^+^ ATPase creating an acidic environment. On the other hand, TNFα activates neutral sphingomyelinase 2 which in turn inactivates H^+^ ATPase leading to a more basic pH within the MVB.

**Figure 3 ijms-23-07837-f003:**
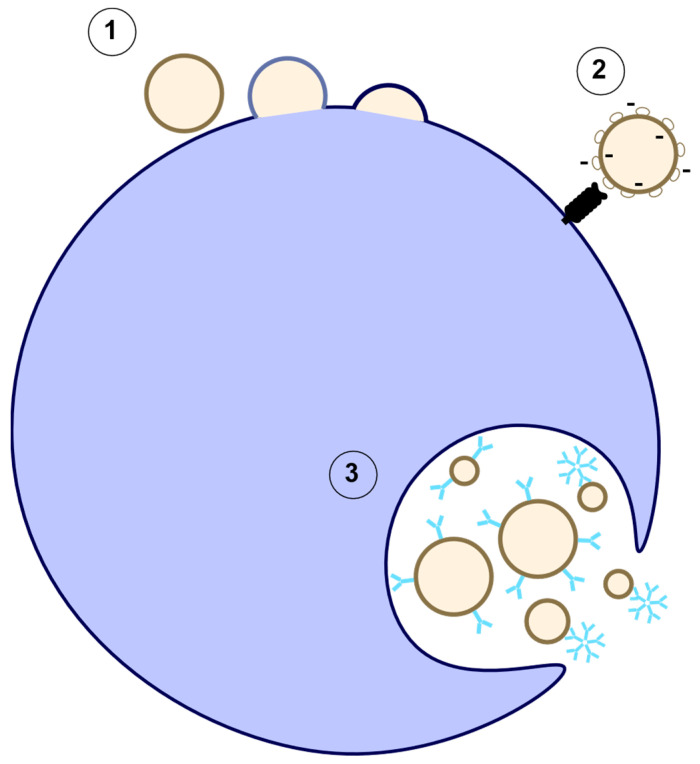
EV Uptake by Cells. EVs can be taken up and cleared from circulation via three methods. (1) The EV membrane can fuse with the plasma membrane of the cell, releasing the EV contents into the cytoplasm. (2) Receptor-mediated endocytosis can occur when a receptor on the cell membrane, such as lactadherin, attaches to a molecule on the EV, such as negatively charged phosphatidylserine. (3) EVs can be endocytosed or phagocytosed. When EVs are coated with molecules like immunoglobulins, phagocytosis of the EVs is increased. When coated with IgG, EVs are endocytosed to a greater degree regardless of their size, however, when they are coated with IgM, phagocytosis is preferential for smaller EVs.

**Figure 4 ijms-23-07837-f004:**
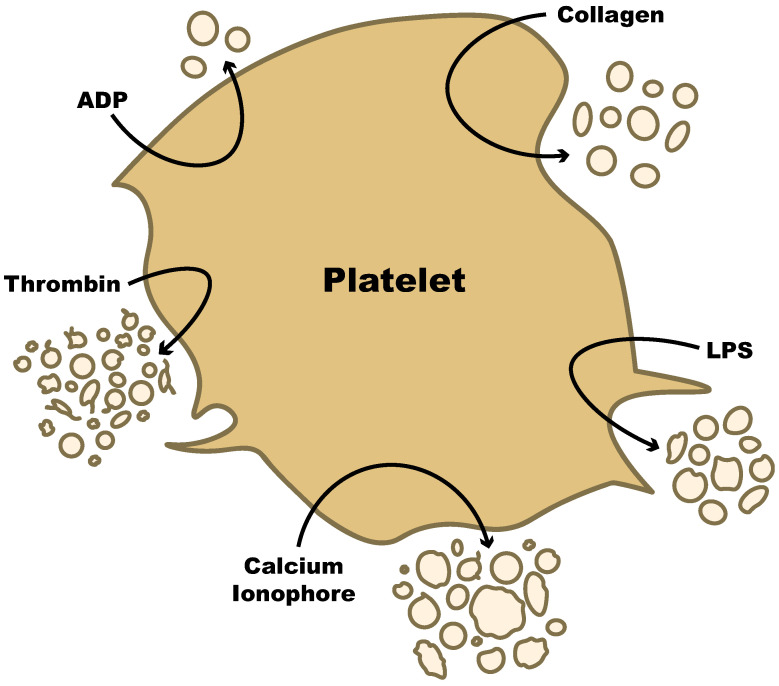
Diverse production of platelet-derived extracellular vesicles with different agonists. Platelets stimulated with different agonists produce extracellular vesicles (EVs) that are characteristically different. ADP and collagen stimulation produces spherical and smooth EVs and a higher number of EVS are produced with collagen. Thrombin stimulation produces an even larger number of EVs, but they are smaller, both spherical and elongated in shape, and have rough surfaces with thin offshoots. Calcium ionophore stimulation produces EVs that are larger and more heterogenous in size and have a rougher membrane surface. Included here is also LPS, a non-host derived stimuli, that leads to PEV production.

**Figure 5 ijms-23-07837-f005:**
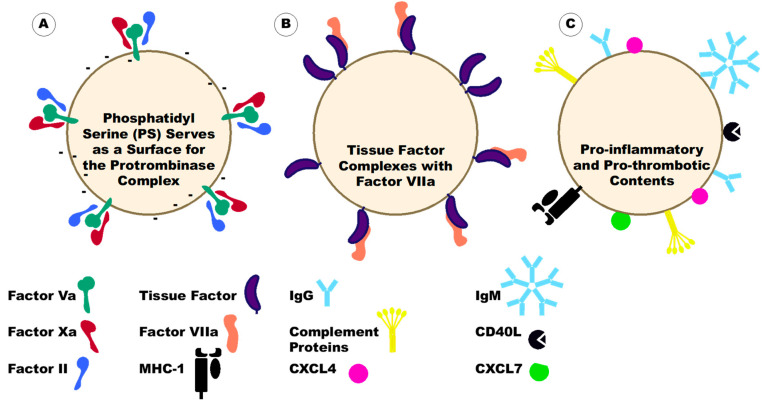
Mechanisms by which Extracellular Vesicles (EVs) can Contribute to Thrombosis. There are three known pathways through which EVs can influence thrombosis: (**A**). Phosphatidylserine on the surface of EVs provides a negatively charged surface on which the prothrombinase complex (comprised of factors Va, Xa, and II) can form, (**B**). Tissue factor (TF) on EVs interacts with factor VIIa to initiate the coagulation cascade, (**C**). Molecules such as clotting factors, MHC-1, complement proteins, CXCL4, CXCL7, CD40 ligand (CD40L), and immunoglobulins (IgG and IgM) on the vesicle membrane can interact with other cell types to produce a pro-inflammatory and pro-thrombotic environment.

**Figure 6 ijms-23-07837-f006:**
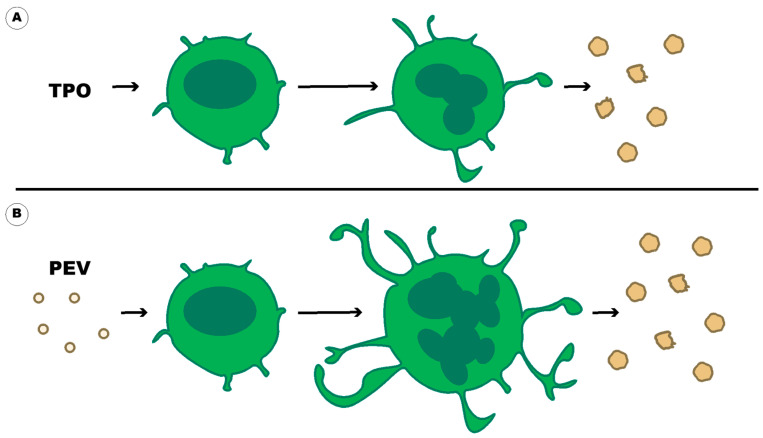
Platelet Extracellular Vesicles (PEVs) Stimulate Megakaryocyte Maturation and Platelet Production. (**A**). Megakaryopoiesis is normally induced by thrombopoietin (TPO), where megakaryocytes become polyploid and platelets are produced from pseudopodia extending from the mature megakaryocyte. (**B**). PEVs can stimulate megakaryopoiesis independent of TPO. When PEVs stimulate megakaryocyte maturation and platelet production, the resulting megakaryocytes are larger and more polyploid and as a result, there is increased platelet production.

**Table 1 ijms-23-07837-t001:** Cell-type specific markers of EVs.

Cell of Origin	Protein Marker	References
Platelet	CD41, CD42b, CD61, CD62P	[2,3,5,9,37,52,53,54,55,56,57,58,59,60,61,62,63,64,65,66,67,68,69,70,71]
Megakaryocyte	CD41, CD42b, filamin A	[2,5,6,37]
Red Blood Cell	Glycophorin A (CD235)	[5,7,58,66,69,70,71]
Endothelial Cell	CD62E, CD31, CD105, CD144, CD146	[5,53,57,59,60,61,62,64,67,68,69,70,71,72,73,74,75,76,77]
Neutrophil	CD16, CD66b	[3,6,63]
Monocyte	CD14, CD11b	[3,5,6,9,53,58,61,70,71,76,78]
Leukocyte	CD45	[5,6,59,61,67,68,71,72,76,79]
T-Cell	CD3, CD4, CD8	[3,5,9,62]
B-Cell	CD19, CD20, CD79a	[3,5]
NK Cell	CD56	[67,80,81,82]
Prothrombotic	CD142 (tissue factor)	[10,39,54,83]

## Data Availability

Not applicable.

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
