# Peer review of "The Role of Platelet-Derived Extracellular Vesicles in Immune-Mediated Thrombosis"

_ijms, 2022, doi:10.3390/ijms23147837_

Round 1

Reviewer 1 Report

This excellent, relatively novel manuscript reviews the role of platelet-derived extracellular vesicles in immune-mediated thrombosis. The scope of the review is broad and in-depth, and is extremely well and clearly written. The cited literature is extensive and appears balanced. However, it suffers a number of grammatical errors and some other minor weaknesses in its current form.

  1. The main weakness is the lack of figures, with only a relatively simply figure presented. The authors could consider further figures and/or a more detailed Figure 1.
  2. Title: "Immune Mediated" should be "Immune-mediated".
  3. Abstract: "3 major" should be "three major"
  4. In a number of places a hyphen is absent, failing to link terms, e.g. "ceramide regulated" (line 43). Please correct and check entire manuscript for other instances.
  5. "+" (and "-") could be superscripted but more so "+" and "-" are not used consistently with "positive" and negative" used instead, for e.g. "annexin V" (e.g. line 179). Please correct and check entire manuscript for other instances.
  6. "ESCRT" (line 56), "ALIX" (line 61), "HUVEC" (line 192), "VWF" (line 222), "NOX1" (line 268), "ESR" and "CRP" (line 464), "PBMCs" (line 658), "LDH" (line 607), "DIC" (line 612), "ARDS" (line 614)", "ACE2" (line 650) should be defined in full. Check all abbreviations are defined on first use.
  7. Insert period after "[34]" (line 82).
  8. In many places unnecessary upper case is used, when lower case would suffice, for e.g. "Cytocholasin" (line 83), "Okadaic Acid" and Calyculin A" (line 88), "Filamin A" (line 99), "Natural Killer (line 141ff), "Mesnchymal Stem" (line 141), "Lactadherin" (line 179ff), "Cyclohexamide" (line 750). Please correct and check entire manuscript for other instances.
  9. "Phosphatidyl serine" should be "Phosphatidylserine" (line 115).
  10. Check Table format (including font) complies with journal and correct accordingly. Standardize terms in Column 1, so that all terms are plural or singular. Correct "NK-Cell" to "NK cell" and "Tissue Factor" to "tissue factor".
  11. Adjust "222.6" to "223" for consistency (line 162).
  12. In numerous places (e.g. line 169), there appears to be at least two spaces (not one). Use find and replace function of Word to correct throughout.
  13. Change "two cells" to "two cell types" (lines 203).
  14. Change "minutes" to "min" for consistency (line 207).
  15. Change "calcium Ionophore" to "calcium ionophore" (line 265).
  16. Numerals with "fold" not consistent (e.g. line 308 vs. 409). Correct and check throughout.
  17. "ANCA" should be "ANCAs" for consistency (line 460).
  18. Correct "p-cadherin" to "P-cadherin" (line 475) here and elsewhere.
  19. Arguably "thrombi" should be used in place of "clots" (line 524) when describing in vivo studies.
  20. Correct "rac1" to "Rac1" (line 544).
  21. Insert space between "receptor[118]" (line 587).
  22. Use proper scientific notation for numerals (lines 597 and 598).
  23. Correct "AV" to annexin V" (line 709) and check entire manuscript.
  24. Correct "regime" to "regimes" (line 731).
  25. Please check and clarify the sentence "The addition of Cyclohexamide..." (line 750ff); as Cyclohexamide impairs protein synthesis, so how does this relate to "this phenomenon"?
  26. Insert reference to support sentence "This is important..." (lines 795-6).
  27. Not necessarily for inclusion in the closing remarks or even in the manuscript, but could PEVs be involved in the rare clotting events associated with some COVID-19 vaccines or have they been studied?
  28. The references contain errors especially in journal titles, for e.g. BR. J. Haematol., JTH, JBC, ATVB. also "=" in Ref. 86 author. Please check and correct throughout.
  29. Delete blank page (page 27).

Author Response

We would like to thank the reviewer for their thoughts and careful review of our paper. We have gone through the text and fixed typographical errors, and formatting errors with references and have added new figures. Below we prepare a point-by-point response to the comments.

Reviewer 1

This excellent, relatively novel manuscript reviews the role of platelet-derived extracellular vesicles in immune-mediated thrombosis. The scope of the review is broad and in-depth and is extremely well and clearly written. The cited literature is extensive and appears balanced. However, it suffers a number of grammatical errors and some other minor weaknesses in its current form.

  1. The main weakness is the lack of figures, with only a relatively simply figure presented. The authors could consider further figures and/or a more detailed Figure 1.

Response: We have added 5 additional figures to the paper.

  1. Title: "Immune Mediated" should be "Immune-mediated".

Response: This has been changed.

  1. Abstract: "3 major" should be "three major."

Response: This has been changed (line 9)

  1. In a number of places, a hyphen is absent, failing to link terms, e.g. "ceramide regulated" (line 43). Please correct and check entire manuscript for other instances.

Response This has been changed throughout the paper. And ceramide- regulated is now in line 60.

  1. "+" (and "-") could be superscripted but more so "+" and "-" are not used consistently with "positive" and negative" used instead, for e.g. "annexin V" (e.g. line 179). Please correct and check entire manuscript for other instances.

Response: This has been changed such that the words positive and negative are being used. These changes can now be found on lines 190-192, 593, 666, and 740.

  1. "ESCRT" (line 56), "ALIX" (line 61), "HUVEC" (line 192), "VWF" (line 222), "NOX1" (line 268), "ESR" and "CRP" (line 464), "PBMCs" (line 658), "LDH" (line 607), "DIC" (line 612), "ARDS" (line 614)", "ACE2" (line 650) should be defined in full. Check all abbreviations are defined on first use.

Response: Thanks for pointing out this error. These changes are now reflected in line 71, 77, 225, 268, 315, 530, 586, 635, 685, 692, 729 and throughout the manuscript.

  1. Insert period after "[34]" (line 82).

Response This has been changed and the reference is, see line 109.

  1. In many places unnecessary upper case is used, when lower case would suffice, for e.g. "Cytocholasin" (line 83), "Okadaic Acid" and Calyculin A" (line 88), "Filamin A" (line 99), "Natural Killer (line 141ff), "Mesnchymal Stem" (line 141), "Lactadherin" (line 179ff), "Cyclohexamide" (line 750). Please correct and check entire manuscript for other instances.

Response: This has been changed in lines 110, 115, 125, 171, 212-213, 838 and throughout the manuscript.

  1. "Phosphatidyl serine" should be "Phosphatidylserine" (line 115).

Response: This has been changed (line 143).

  1. Check Table format (including font) complies with journal and correct accordingly. Standardize terms in Column 1, so that all terms are plural or singular. Correct "NK-Cell" to "NK cell" and "Tissue Factor" to "tissue factor".

Response: The font in the table has been changed and the cell types are all consistently singular and the tissue factor was uncapitalized.

  1. Adjust "222.6" to "223" for consistency (line 162).

Response: This has been changed (line 193).

  1. In numerous places (e.g. line 169), there appears to be at least two spaces (not one). Use find and replace function of Word to correct throughout.

Response: We have looked through the manuscript and changed the two one space where we found this problem.

  1. Change "two cells" to "two cell types" (lines 203).

Response: This has been changed (line 248).

  1. Change "minutes" to "min" for consistency (line 207).

Response: To be consistent throughout the text we have changed all min to minutes (lines 266 and 267).

  1. Change "calcium Ionophore" to "calcium ionophore" (line 265).

Response: This has been changed (line 311).

  1. Numerals with "fold" not consistent (e.g. line 308 vs. 409). Correct and check throughout.

Response: This has been changed (lines 355 and 474).

  1. "ANCA" should be "ANCAs" for consistency (line 460).

Response: This has been changed (line 525).

  1. Correct "p-cadherin" to "P-cadherin" (line 475) here and elsewhere.

Response: This has been changed (line 542).

  1. Arguably "thrombi" should be used in place of "clots" (line 524) when describing in vivo studies.

Response: Whenever in vivo studies were in reference we changed the use of the word clots to thrombi. Changes have been made on lines 420, 421, 422, 427, 463, 466, 595, 671, 689, and 759.

  1. Correct "rac1" to "Rac1" (line 544).

Response:This has been changed (line 616).

  1. Insert space between "receptor[118]" (line 587).

Response: This has been changed (line 665).

  1. Use proper scientific notation for numerals (lines 597 and 598).

Response: This has been changed (line 675-676).

  1. Correct "AV" to annexin V" (line 709) and check entire manuscript.

Response: This has been changed (lines 504 and 788).

  1. Correct "regime" to "regimes" (line 731).

Response: This has been changed (line 811).

  1. Please check and clarify the sentence "The addition of Cyclohexamide..." (line 750ff); as Cyclohexamide impairs protein synthesis, so how does this relate to "this phenomenon"?

Response: Thank you to the reviewer for pointing this out. We had meant to say that protein synthesis was necessary and have changed the text to reflect this (line 838).

  1. Insert reference to support sentence "This is important..." (lines 795-6).

Response: We have added a new reference to help support this sentence (reference 219) and this can be found in line 895.

  1. Not necessarily for inclusion in the closing remarks or even in the manuscript, but could PEVs be involved in the rare clotting events associated with some COVID-19 vaccines or have they been studied?

Response: This is a great idea and we have added information in lines 823-829. To our knowledge, no mechanistic relationship has been identified to date between PEVs and the COVID vaccine related thrombotic events, however there is speculation and a potential mechanism that we have now included.

  1. The references contain errors especially in journal titles, for e.g. BR. J. Haematol., JTH, JBC, ATVB. also "=" in Ref. 86 author. Please check and correct throughout.

Response: We have gone over the references again and made corrections.

  1. Delete blank page (page 27).

Response: This has been changed.

Reviewer 2 Report

The assessed article entitled „The Role of Platelet-Derived Extracellular Vesicles in Immune Mediated Thrombosis” is an interesting review paper that describes the topic in depth. Authors characterize different types of extracellular vesicles (EV), including exosomes, microparticles/microvesicles and apoptotic bodies. Much attention is paid to the development of Platelet-derived EV under several disease conditions, especially in autoimmune or infectious diseases and infections. The mechanism of cellular interactions and pathological role of PEVs are highlighted. Authors describe that PEVs have also a feedback loop where they can act on platelets and megakaryocytes -  their parent cells, which can result in increased formation of pro-thrombotic PEVs. Thus this review paper allows to better understand how PEVs are being generated in different states of disease. The understanding the role of PEVs is a great clinical importance and is necessary in order to provide appropriate treatment to patients. 

I have no major comments on the completeness of the topic covered and the appropriateness of references. There is difficult to find a gaps in the knowledge. However some of issues remain to be explained. 

1) What is the meaning  “all” behind lymphocytes in Table 1? The selection of protein markers to cells of origin is also debatable. For example CD45 is expressed on all leukocytes (not only at lymphocytes as signed in Table 1). The table should be redrafted, so that the data is consistent with the facts.

2) In lines 404-5 is statement that “in lupus, the platelet are smaller and more activated, leading to increased production of PEVs [138” – but the appropriate reference should be 139], which is misinterpretation. In the general population it is thought that larger platelets are more active and contribute to increased risk of cardiovascular disease. In active inflammatory conditions, large and activated platelets are consumed preferentially at the site of inflammation, leaving small platelets behind, what may explain lower platelets size (MPV) in systemic lupus patients. Also an increased amounts of small platelet microparticles may contribute to lower MPV values of platelets. In fact, decreased platelet size in lupus and associated with microparticle formation and  antiphospholipid syndrome could participate in enhanced coagulability. However, the claim that small platelets contribute to the increased formation of platelet microparticles seems somewhat illogical, because others mechanisms likely contribute in this process.

3) References starts from [2], because as [1] is a title of this paragraph. Therefore, probably some of the references are badly associated with the described topics.

4) My recommendation is that the authors should have carefully checked text to eliminate some typographical errors.

The reviewed paper is merit, relevant and could be of interest to scientific community. It is written in clear and simple language – easy to interpret and understand. The cited references are relevant (almost half of them have been published in last 5 years), but some of them are connected wrongly (probably due the citation [1]). In my opinion, after minor revision, this paper could be considered worthy of publication. 

Author Response

We would like to thank the reviewers for their thoughts and careful review of our paper. We have gone through the text and fixed typographical errors, and all the formatting errors with references and have added new figures. Below we prepare a point-by-point response to the comments.

Reviewer 2

I have no major comments on the completeness of the topic covered and the appropriateness of references. There is difficult to find a gaps in the knowledge. However, some of issues remain to be explained. 

1) What is the meaning  “all” behind lymphocytes in Table 1? The selection of protein markers to cells of origin is also debatable. For example CD45 is expressed on all leukocytes (not only at lymphocytes as signed in Table 1). The table should be redrafted, so that the data is consistent with the facts.

Response: We have redrafted the table and added references to support the findings and have updated the correct information regarding specific markers to use for specific cell type.

2) In lines 404-5 is statement that “in lupus, the platelet are smaller and more activated, leading to increased production of PEVs [138” – but the appropriate reference should be 139], which is misinterpretation. In the general population it is thought that larger platelets are more active and contribute to increased risk of cardiovascular disease. In active inflammatory conditions, large and activated platelets are consumed preferentially at the site of inflammation, leaving small platelets behind, what may explain lower platelets size (MPV) in systemic lupus patients. Also an increased amounts of small platelet microparticles may contribute to lower MPV values of platelets. In fact, decreased platelet size in lupus and associated with microparticle formation and  antiphospholipid syndrome could participate in enhanced coagulability. However, the claim that small platelets contribute to the increased formation of platelet microparticles seems somewhat illogical, because others mechanisms likely contribute in this process.

Response: The reviewer is correct in that larger platelets are typically more active. As such we have gone back and updated our wording to say “The prevalence of smaller platelets is most likely a consequence of larger platelets being consumed preferentially, resulting in more PEVs being produced (presumably from the consumed larger platelets)” so that we can correctly interpret the results from this paper. The problem in the reference was the same as was mentioned in the next comment where the reference section was moved by one due to a formatting error. This has been fixed and the correct reference is now indicated. (Lines 469-471).

3) References starts from [2], because as [1] is a title of this paragraph. Therefore, probably some of the references are badly associated with the described topics. 4) My recommendation is that the authors should have carefully checked text to eliminate some typographical errors.

Response: We are grateful for your notice of this formatting error. We have fixed this and gone back and checked that the references and their in-text numbers match.
